# Effects of storage conditions on the microbiota of fecal samples collected from dairy cattle

**Ana S. Jaramillo-Jaramillo**[1]*, **J. T. McClure**[1‡], **Henrik Stryhn**[1], **Kapil Tahlan**[2‡], **Javier Sanchez**[1]

1 Department of Health Management, Atlantic Veterinary College, University of Prince Edward Island, Charlottetown, Prince Edward Island, Canada, 2 Department of Biology, Memorial University of Newfoundland, St. John's, Newfoundland and Labrador, Canada

☯ These authors contributed equally to this work.
‡ JTM and KT also contributed equally to this work.
* asjaramillo@upei.ca

**Data Availability Statement:** All data and analysis files are available in the GitHub repository at https://github.com/anasojaramillo/Effects-of-

## Abstract

Microbiota analyses are key to understanding the bacterial communities within dairy cattle, but the impact of different storage conditions on these analyses remains unclear. This study sought to examine the effects of freezing at -80°C immediately after collection, refrigeration at 4°C for three days and seven days and absolute ethanol preservation on the microbiota diversity of pooled fecal samples from dairy cattle. Examining 16S rRNA gene sequences, alpha (Shannon, Pielou evenness, observed features and Faith PD indices) and beta (Bray-Curtis, βw and Weighted UniFrac) diversity were assessed. The effects of storage conditions on these metrics were evaluated using linear mixed models and PERMANOVA, incorporating the farm as a random effect. Our findings reveal that 7d and E significantly altered the Shannon index, suggesting a change in community composition. Changes in Pielou evenness for 3d and 7d storage when compared to 0d were found, indicating a shift in species evenness. Ethanol preservation impacted both observed features and Faith PD indices. Storage conditions significantly influenced Bray-Curtis, βw, and Weighted UniFrac metrics, indicating changes in community structure. PERMANOVA analysis showed that these storage conditions significantly contributed to microbiota differences compared to immediate freezing. In conclusion, our results demonstrate that while refrigeration for three days had minimal impact, seven days of refrigeration and ethanol preservation significantly altered microbiota analyses. These findings highlight the importance of sample storage considerations in microbiota research.

## Introduction

Studying microbial communities in various environments has traditionally relied on conventional bacterial culture and identification methods [1]. However, many bacterial species resist these methods, leaving a vast portion of microbial diversity unexplored. To overcome this,

storage-conditions-on-the-microbiome-of-fecal-samples-collected-from-dairy-cattle/tree/main.

**Funding:** - Initials of the authors who received each award: JS - Grant numbers awarded to each author: 1 - The full name of each funder: NSERC Discovery Grant (NSERC DG) - URL of each funder website: https://www.nserc-crsng.gc.ca/index_eng.asp - Did the sponsors or funders play any role in the study design, data collection and analysis, decision to publish, or preparation of the manuscript? No The funders had no role in study design, data collection and analysis, decision to publish, or preparation of the manuscript.

**Competing interests:** The authors have declared that no competing interests exist.

researchers use metagenomics, which sequences DNA from various environments, like water, soil, and biological samples, to reveal microbial diversity and taxonomy [2]. Depending on the genes studied, metagenomics can focus on 16S ribosomal genes for microbiota, antimicrobial resistance genes for the resistome, mobile genetic elements for the mobilome, and virulence genes for the virulome [3]. Microbiota analyses specifically use techniques like identification, classification, and clustering of bacterial 16S rRNA gene sequencing datasets to examine microorganisms residing on or within subjects. Such analyses often employ Alpha and Beta diversity measurements. Alpha diversity assesses species diversity within individual samples, including bacterial richness, unique operational taxonomic units (OTUs), phylogenetic diversity, and species evenness. Conversely, Beta diversity measures the average dissimilarity from an individual unit to the group centroid [4, 5].

In dairy and beef production, microbial communities are commonly studied for their roles in animal health, well-being, productivity, and potential public health implications [6, 7]. The fecal and gut microbiotas are particularly interesting due to their significant influence on host physiology, immunity, and metabolism [8, 9]. However, very little is documented about the effect of sample collection, transportation, and storage conditions on fecal microbiota in cattle. Several studies regarding the impact of storage conditions on fecal microbiota have been conducted widely in fecal samples from humans and, to a lesser extent, in samples from ruminants, companion animals, and wild animals [10–13]. Freezing at -20°C or lower (-80°C) is common for fecal sample storage [14–17]. Alternative methods, like immersion in 70%, 95%, or 100% ethanol, have been used due to their ability to preserve DNA at ambient temperatures [18]. However, these methods can degrade sample quality for conventional microbiology culturing and necessitate careful transportation procedures [10, 19].

Variations in storage conditions may alter the microbiota, making it challenging to compare results across studies. Ensuring optimal storage conditions is vital, particularly when processing is delayed or when samples are collected from remote locations, such as rural dairy farms. Any delay in processing could distort the microbiota analysis results [14, 15]. This study aims to establish practical strategies for storing and transporting pooled fecal samples collected from dairy cattle for microbiota analysis. We investigated the effects of different preservation methods on microbiota diversity, including immediate freezing of samples at -80°C upon collection, refrigeration at 4°C for three and seven days, and ethanol preservation at room temperature. To our knowledge, this is the first study investigating the effect of the mentioned conditions in fecal samples from dairy cattle. We hypothesized that refrigerating the samples at 4°C for a shirt period (one to three days) could be a convenient method for sample transportation and storage when freezing is impossible.

## Materials and methods

### Ethics statement

Owners of the farms selected for conducting this work gave their permission to carry out the study. No additional permissions for sample collection and analysis were needed.

### Study design and sample collection

For this observational cross-sectional study, 28 dairy farms from Prince Edward Island (PEI) and 21 from Nova Scotia (NS), Canada, enrolled in a research project on antimicrobial stewardship and resistance in dairy cattle, were visited from July to December 2020. Inclusion criteria included a minimum herd size of 50 lactating cows had to be on the DHI (Herd Management Score) program [20] and raise their own heifers. Participating farms could include free-stall or tie-stall housing. During the milking time, approximately 100 g of feces

was collected directly from the rectum of five lactating cows. All individual fecal samples were labelled with the cow's ID, respectively, and one sample set per farm was obtained and transported to the AVC laboratory in coolers with ice packs, ensuring a temperature not exceeding 8˚C. The average time from collection to arrival at the laboratory was 3 hours for samples from PEI farms and 24 hours for samples from NS. The varying times and potential temperature fluctuations during sample transportation were limitations of this study. To mitigate biases, we included the farm as a random effect in all statistical models.

**Sample selection and processing.** The selection of samples was made by convenience based on their arrival at the AVC laboratory. The first 22 sample sets (11 from PEI and 11 from NS) that arrived at the laboratory were included and processed in this study. Aliquots of 20 grams from each farm's individual cow fecal samples were pooled and homogenized manually using a spatula to get one pooled sample per farm. Then, the resulting pooled samples were divided into four subsamples of 5 grams each (n = 88 subsamples in total). One subsample was stored at -80˚C freezer upon arrival at the laboratory (0d) until DNA extraction, another subsample was held at 4˚C for three days (3d) and then stored at -80C until DNA extraction, and another subsample was held at 4˚C for seven days (7d). The last subsample was mixed with 5 ml (1:1) of ethanol 100% (E) in a 15 ml conical tube and kept at room temperature (RT) until DNA extraction. All 4 subsamples were processed for DNA extraction on day seven after arrival at the laboratory.

## DNA extraction

DNA extractions for all subsamples were carried out using the Qiagen PowerMax Soil Kit (Qiagen Laboratories) according to the manufacturer's instructions. Frozen fecal samples were thawed overnight at 4˚C, and 0.25 grams from each subsample was used for the extraction. The DNA yield and quality were checked using the NanoDrop 1000 Spectrophotometer (Thermo Fischer Scientific). Negative or positive controls were not included during the DNA extraction. A sample was re-extracted if it did not achieve a DNA yield of more than 30 ng/μL, a 260/280 ratio of over 1.80, and a 260/230 ratio of over 2.0. The extracted DNA was then stored at -80˚C until it was shipped for sequencing.

## 16S rRNA gene sequencing and data processing

For microbiota sequencing, 100 μL of extracted DNA from each subsample was sent to the Integrated Microbiota Resource (IMR) laboratory (Dalhousie University, Halifax, NS, Canada) for 16S rRNA gene amplification and sequencing. The V6-V8 16S subunit bacteria-specific region was amplified with the primer set [5'–ACGCGHNRAACCTTACC–3'] / [5'–ACGG GCRGTGWGTRCAA–3'] to yield approximately 400–500 bp DNA fragments. Amplicon sequencing was performed on the Illumina MiSeq instrument (Illumina, San Diego, CA, USA) to produce $2 \times 300$ bp paired-end read lengths with up to 100,000 reads per sample. The sequenced 16S rRNA amplicons were analyzed using Quantitative Insights Into Microbial Ecology version 2 (Qiime2-2020.11) tools [21]. Briefly, all reads were processed for sequence quality and denoising using DADA2 [22]; meanwhile, chloroplast and mitochondrial DNA contaminants were removed from the dataset. All data and analysis files are available in our GitHub repository at the following link: https://github.com/anasojaramillo/Effects-of-storage-conditions-on-the-microbiome-offecal-samples-collected-from-dairy-cattle/tree/main.

## Taxonomy, diversity indices calculation and statistical analysis

Operational Taxonomic Units (OTUs) were assigned using a naive Bayes classifier trained on the Greengenes database [23] and were classified at phylum, class, order, family, and genus

levels. Relative abundances of dominant bacterial populations under different storage conditions were calculated at the phylum level.

Alpha diversity (Shannon, observed features, Pielou evenness, and Faith PD) was determined, and the effect of storage conditions on these metrics was compared. Statistical analyses of the four alpha diversity indices were performed using general linear mixed models. Pielou evenness, observed features, and Faith PD were power transformed based on a Box-Cox analysis [24] to meet the assumptions of normality and homoscedasticity. One model was constructed for each index, including the storage conditions (0d, 3d, 7d and E) as a fixed effect and the farm as a random effect. Six pairwise comparisons were carried out after a Bonferroni adjustment.

Beta diversity distances (Bray-Curtis, βw and Weighted UniFrac) were used to conduct Principal coordinates analysis (PCoA) and Permutational multivariate analysis of variance (PERMANOVA) to visualize patterns in the community structure between samples. One PERMANOVA model was constructed for each distance matrix, including the storage conditions as a fixed effect and the farm as a random effect. Also, the distances in each matrix were grouped by pair of storage conditions considering the farm effect on these differences.

Alpha diversity, beta diversity and taxonomy relative frequencies were calculated at a rarefaction depth of 9400 seq/sample using the Phyloseq [25] and the DiversitySeq [26] packages in R version 4.0.4 (R Core Team, 2021]. All statistical analysis was performed with Stata 16.1 (StataCorp, College Station, Texas, USA) software, except that the R package vegan (version 4.0.4) was used for PERMANOVA. The figures were performed with GraphPad Prism 9 (v9.5.0) (GraphPad Software, San Diego, California) and the R package ggplot2 (version 4.0.4).

## Results

### Sample processing

Among the 88 DNA samples sent for sequencing, 17 failed the sequencing QC (1, 3, 5, and 8 from the 0d, 3d, 7d, and E groups, respectively). For the remaining 71 samples, the average number of reads per sample and the reads range after post-trimming and alignment are shown in Table 1.

### Alpha diversity

A significant effect of storage conditions was observed on alpha diversity indices. For the Shannon index (Fig 1A), the 7d and E storage conditions were found to reduce microbial richness compared to the 0d condition. The Pielou evenness index also displayed variations under different storage conditions, decreasing in the 3d and 7d refrigeration conditions compared to the 0d condition. In contrast, the ethanol condition (E) increased the evenness compared to

**Table 1. Sequencing summary by storage condition.**

| Condition | Sequenced samples | Reads per Sample | Range |
|:---:|:---:|:---:|:---:|
| 0d | 21 | 16,272.8 | 5,615–29,465 |
| 3d | 19 | 17,518.0 | 7,169–27,904 |
| 7d | 17 | 17,296.0 | 7,031–38,528 |
| E | 14 | 13,403.0 | 4,891–24,254 |

Number of sequenced samples, the average reads per sample, and the range of reads for each storage condition. The conditions include immediate freezing at -80˚C (0d), refrigeration at 4˚C for three days (3d) and seven days (7d), and ethanol preservation at room temperature (E).

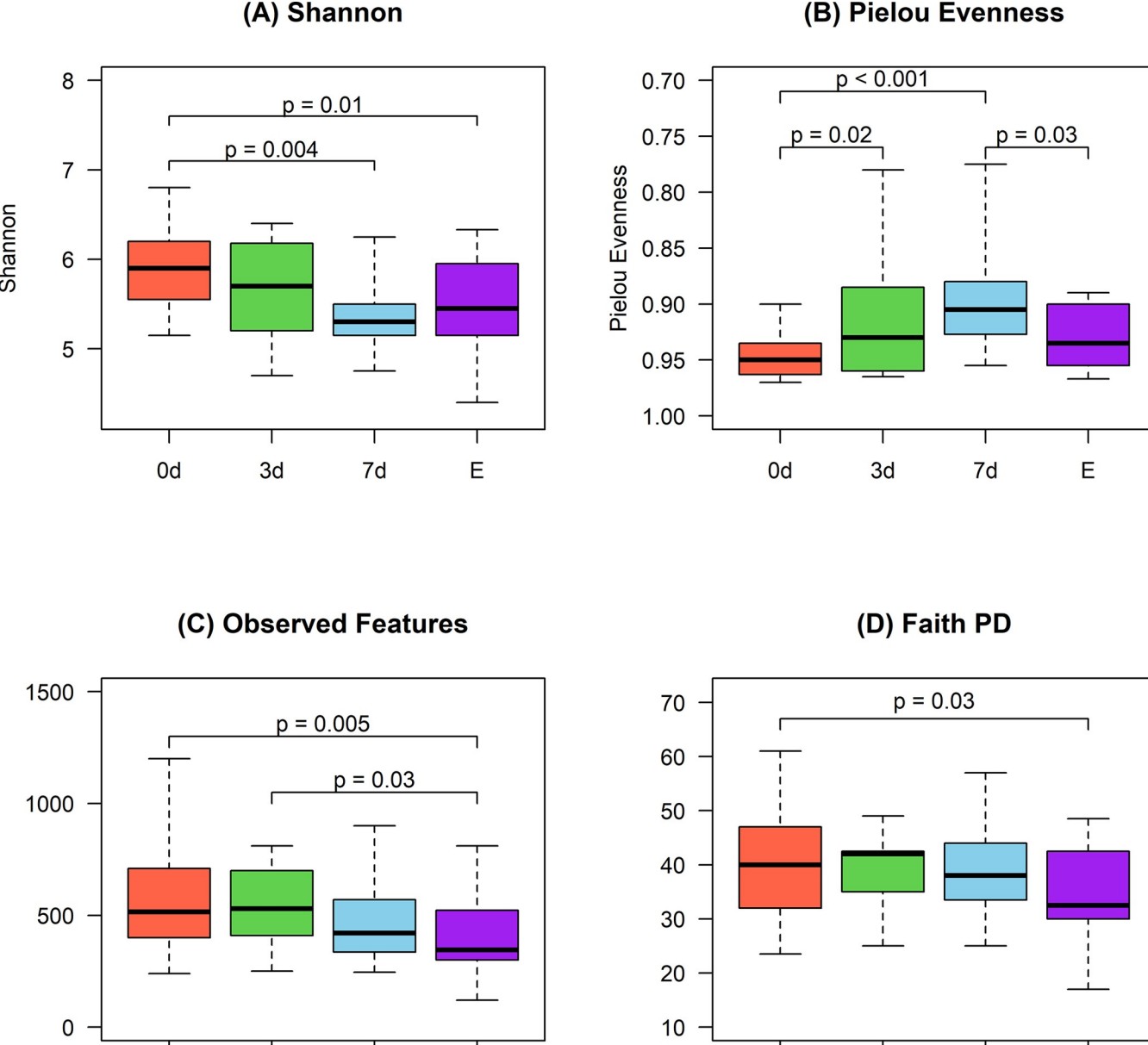

**Fig 1.** Box plots representing alpha diversity indices: (A) Shannon, (B) Pielou Evenness, (C) Observed features, and (D) Faith PD, under four different storage conditions: Immediate freezing at -80°C (0d), refrigeration for three (3d) and seven (7d) days at 4°C, and preservation in ethanol (E). For better visualization, the y-axis scale in plot B was adjusted. Significant differences (p ≤ 0.05) between methods are indicated by p-values shown on a solid line.

the 7d group (Fig 1B). The Observed features abundance (Fig 1C) and the Faith PD index (Fig 1D) were notably affected by the E storage condition compared to the 0d condition. Furthermore, the E condition reduced observed features compared to the 3d condition.

## Beta diversity

For beta diversity, the median distances between six pairs of four different storage conditions were summarized in a boxplot (Fig 2). Compared to E, all other storage conditions showed

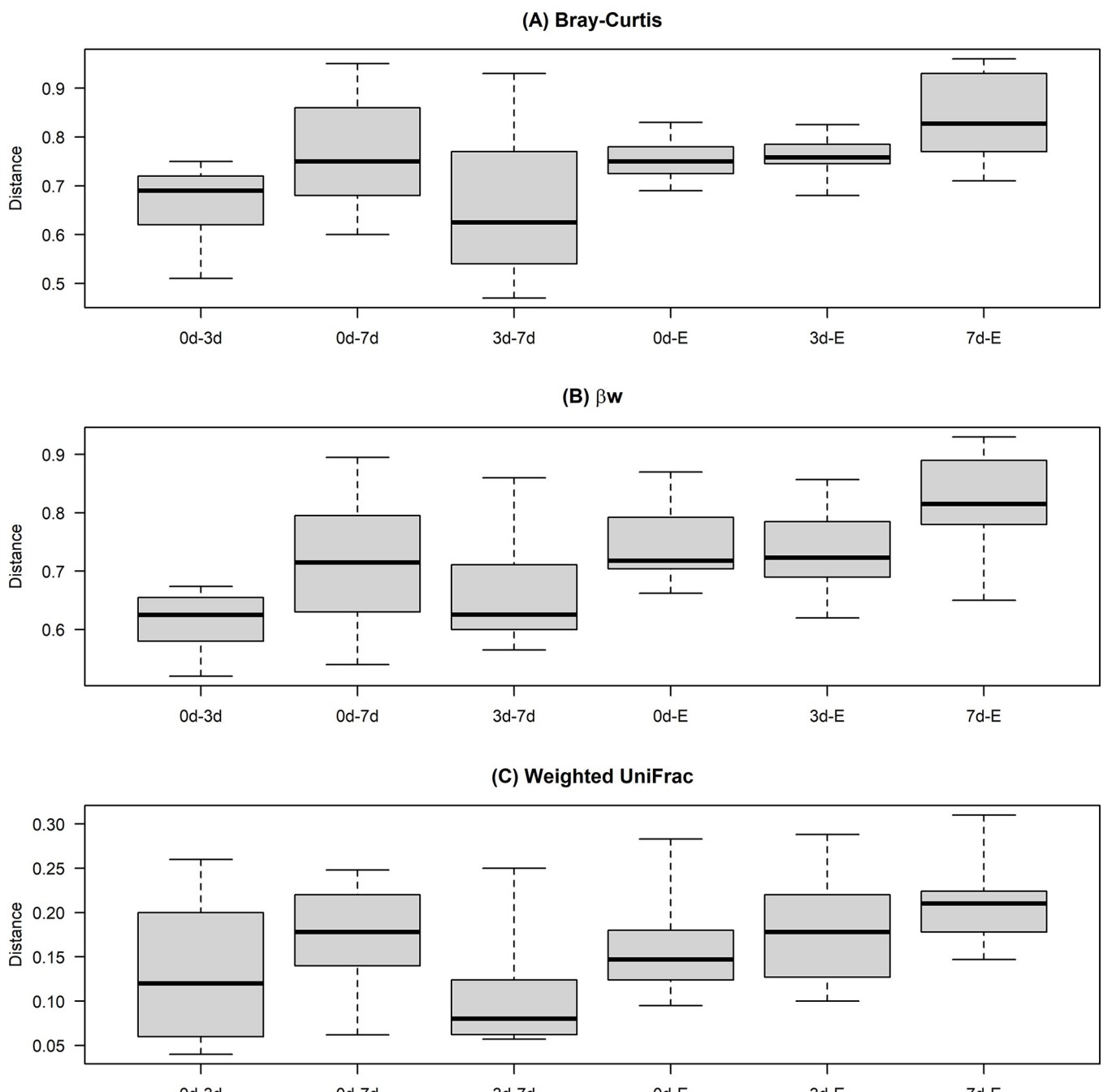

**Fig 2.** Box plots of beta diversity (A) Bray-Curtis (B) βw and (C) Weighted UniFrac for six pairs of four storage conditions: Immediate freezing at -80˚C (0d), refrigeration for three (3d) and seven (7d) days at 4˚C, and preservation in ethanol (E).

more pronounced differences in the microbial communities with less variability pattern observed in the three beta diversity measurements. On the other hand, when comparing both refrigeration methods, median distances tended to be the smallest but with large variability, especially in the Bray-Curtis diversity measurement.

PERMANOVA test analysis on the Bray-Curtis, βw and Weighted UniFrac beta diversity indicated that the storage conditions contributed significantly to the differences in the

**Table 2. PERMANOVA test analysis on beta diversity distances.**

| Beta Distance | Method (3 df) | | Farm ID (21df) | |
|---|---|---|---|---|
| | R2 | P | R2 | P |
| **Bray-Curtis** | 0.12 | 0.001 | 0.40 | 0.001 |
| **βw** | 0.09 | 0.001 | 0.42 | 0.001 |
| **Weighted UniFrac** | 0.29 | 0.001 | 0.38 | 0.001 |

PERMANOVA test analysis on the Bray-Curtis, βw and Weighted UniFrac beta diversity of the samples exposed at four conditions. df = degree of freedom; R2 = R square and P = P-value.

microbial composition of the samples ($p < 0.001$) but only explained 12%, 9% and 29% of the total variation in the data, respectively. On the other hand, farm variability accounted for 40%, 42%, and 38% of the total variation, respectively (Table 2).

The Bray-Curtis and Weighted UniFrac PCoA plots, in Fig 3A and 3C, samples from the 0d storage mostly clustered together, somewhat overlapping with the 3d samples but less with the 7d. Consistent across the three plots, some E and 7d samples stand out distinctly. Each plot also indicates how much variation is explained by its main components.

## Taxonomy

For bacterial composition, the relative abundances of the dominant phyla were calculated and are shown in Fig 4. At the phylum level, the samples from all storage conditions shared common phyla, with the most prevalent being Bacteroidetes, Firmicutes, and Proteobacteria. Compared to the 0d condition, the relative abundance of Bacteroidetes decreased in the 3d, 7d, and E conditions. Conversely, the relative abundance of Firmicutes decreased in the 3d and 7d conditions but increased in the E condition compared to 0d. Additionally, the relative abundance of Proteobacteria increased in the 3d and 7d conditions. These observations suggest that storage duration and conditions impact the bacterial composition at the phylum level.

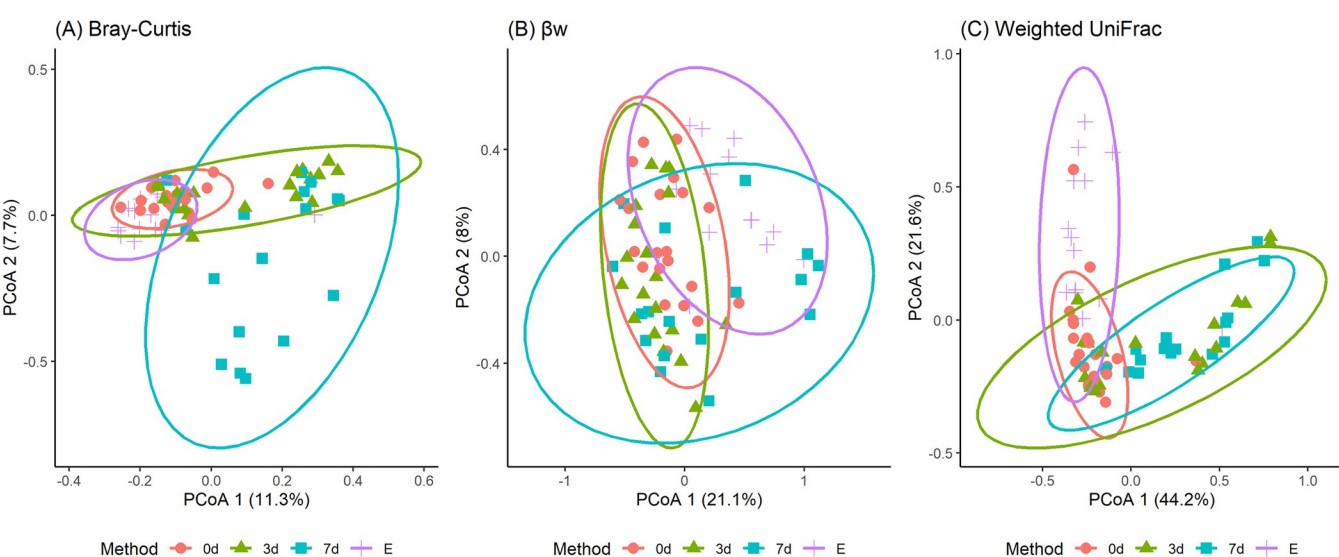

**Fig 3. Clustering of samples due to storage conditions by PCoA.** Ordination plots of (A) Bray-Curtis, (B) βw and (C) Weighted UniFrac beta diversity measured for four storage conditions: Immediate freezing at -80˚C (0d), refrigeration for three (3d) and seven (7d) days at 4˚C, or preservation in ethanol (E). The ellipses indicate the clusters by groups, showing the 95% confidence intervals.

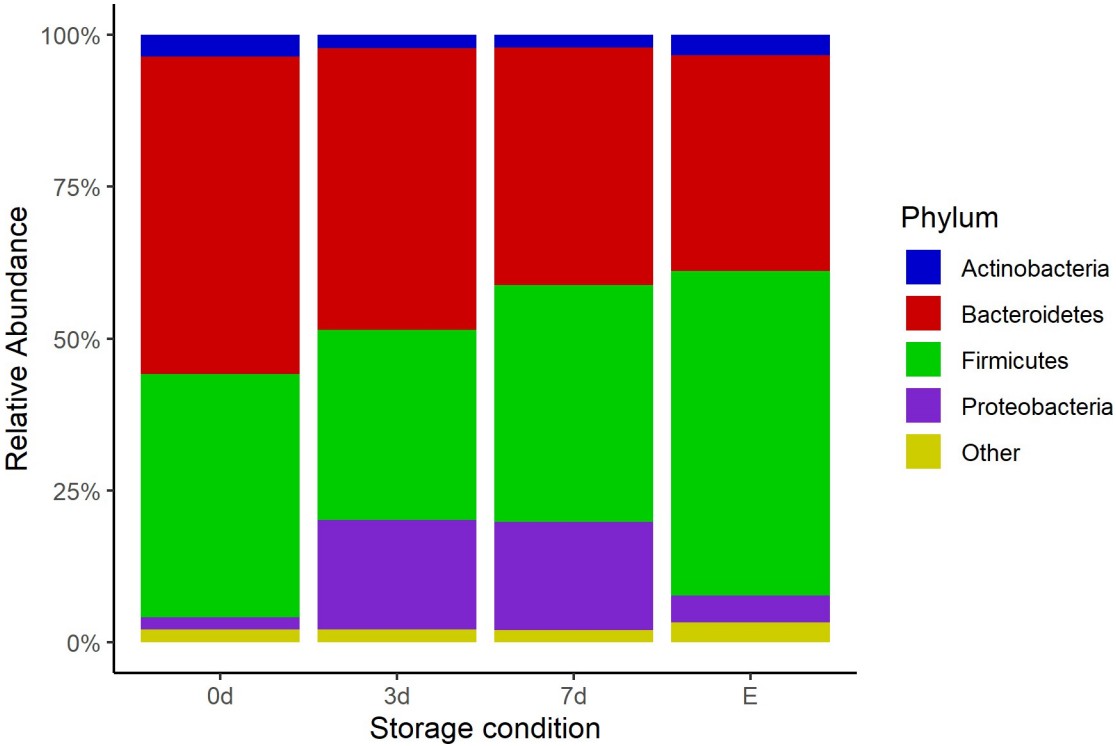

**Fig 4. Bar plot of relative abundance at phylum level for samples storage under four different conditions.** Immediate freezing at -80˚C (0d), refrigeration for three (3d) and seven (7d) days at 4˚C, or preservation in ethanol (E).

## Discussion

Storing and transporting fecal samples collected for microbiota analysis may implicate differences in diversity measures such as the richness and evenness of taxonomic groups and taxonomy frequencies. Variations in these factors may alter the microbiota, masking or exacerbating some results and making it challenging to make inferences and comparisons across groups of interest or studies. This is especially a concern when the samples' specific microbial features or characteristics are the main topic of interest [27].

Freezing at -20˚C or lower (-80˚C) is the recommended practice for fecal sample storage [9–12]. Furthermore, preservative solutions, like 95% or 100% ethanol and refrigeration at 4˚C, are commonly used when freezing the samples is not possible [13]. However, in some cases, these methods necessitate careful transportation procedures to ensure the optimum conditions of the samples [5, 14]. The present study investigated the effects of refrigeration at 4˚C for three and seven days and ethanol preservation at room temperature on microbiota diversity, compared to the immediate freezing of samples at -80˚C upon collection. No previous studies have investigated these methods' effects on fecal samples from cattle.

Alpha diversity results indicate that samples stored at 7d had a lower richness and evenness than samples stored at 0d, probably due to the several days of exposure to oxygen and specific temperature and light conditions, which may affect the growth of some bacteria positively or negatively. Previous studies in canines, where the temperature and time of storage of fecal samples were investigated, suggested that at 4˚C, samples exhibited a reduction in species richness levels at seven days compared to fresh samples [11]. On the other hand, other studies in humans evaluated the effect on the alpha diversity of fecal samples refrigeration for a storage period from 3 to 72 hours, and no statistical differences were found compared to the -80˚C

frozen or fresh samples, similarly to the present study results [8, 28]. The 100% ethanol conservation method used in the present study decreased the richness, evenness, phylogenetic diversity, and number of observed taxonomic units compared with the 0d method. Previous studies have shown similar results in the alpha diversity of fecal samples stored in 70, 95, or 100% ethanol. Ma et al. showed a bacterial alpha diversity alteration in samples stored for 1 week using 70% ethanol, characterized by significantly decreased Shannon diversity index and number of OTUs compared with fresh samples [29]. It is suggested that 70% ethanol performed poorly as a stabilizing condition for microbial communities, like using no preservative [10, 28]. At 70% concentration, ethanol, a potent solvent, may disrupt bacterial cell membranes, causing cell lysis and releasing enzymes that degrade cellular components, thus failing to preserve microbial communities' original structure and composition effectively [10].

The difference of E storage versus freezing at 0d in this study is similar to the difference in the alpha diversity Shannon found in a recent study where human and canine fecal samples were stored in 70% and 95% ethanol, with 70% ethanol being the only method showing significant differences when compared to freezing at -20˚C [30]. In this study, the average number of observed features in each sample (richness) and the Faith PD (phylogenetic diversity) showed differences in samples stored in E compared to freezing at -80˚C. Similar results were found in a study by Sinha et al. in 2016, where human fecal sample aliquots were preserved in 3 different stabilization solutions, including 70% ethanol and made comparisons to -80˚C freezing. The results showed that the observed features of 70% ethanol preservation samples decreased [31].

For all three beta diversity measurements, the median distance between samples stored 3d and 7d was the lowest of all pairs' but with large variability, which means that some of those refrigerated samples had a similar OTUs abundance, whereas others had a dissimilar OTUs abundance. Furthermore, PCoA analysis and ordination plots using beta diversity showed how samples stored for 7 days at 4˚C tended to cluster far from the other conditions groups and far from each other, while the samples refrigerated for 3 days were still close to the frozen samples. The former could be due to the variable effect of the temperature, oxygen, and time of exposure of the fecal samples during the storage. Our results agree with a previous study where the impact of 3 and 7 days of refrigeration of human fecal samples stored at 4˚C without preservative on beta diversity weighted UniFrac distance was evaluated and resulted significantly larger than the reference (-80˚C) and had a higher variability than the Cary Blair preservation method. Refrigeration is a popular preservation condition for fecal samples, not only for storage purposes but for shipping and transportation because of its low cost; however, it can be suggested that periods of refrigeration longer than three days can affect the bacterial community composition of the fecal microbiota specifically in humans [32].

The median distances between samples stored in E and samples of the other methods showed a less similar community composition and OTUs abundance but with less variability, which could be interpreted as a homogeneous effect of the ethanol on the beta diversity. PCoA ordination plots showed the E samples clustering close to the 0d samples, especially in the Bray Curtis beta diversity. Previous studies have suggested that 95% ethanol or higher can preserve the bacterial DNA better than 70% ethanol in fecal samples stored for long periods, and when freezing the samples is not possible [5].

Additionally, The PERMANOVA analysis showed that 7 days of refrigeration had a larger effect on the beta diversity matrices versus the other storage conditions; however, the farm variability explained a higher proportion of the variation in the data, suggesting an important individual effect on microbial diversity as reported in several other studies [8, 30, 33].

The taxonomy classification in the present study showed that all samples, regardless of the storage condition, shared the same phylum with some changes in the proportions of the most prevalent bacteria. Refrigeration for 3 days decreased the Bacteroidetes and Firmicutes

proportion, without modification of the ratio of these phyla, and increased the Proteobacteria proportion; however, when the refrigeration was up to 7 days, the Bacteroidetes/Firmicutes ratio was affected due to a reduction of the Bacteroidetes bacteria. These changes can be related to the temperature effect and the oxygen levels during refrigeration, decreasing the obligated anaerobes and increasing the facultative anaerobes such as Proteobacteria [34]. Ethanol as a preservation method has been described and utilized commonly due to its protective effect on the bacterial DNA in different samples. Some studies explained that 95% ethanol is sufficient to replace the sample water with ethanol and thereby avoid DNA degradation; however, other studies agree with our results, showing the effect of ethanol in phylum Firmicutes and Proteobacteria [8, 34].

The absence of negative and positive controls in the DNA extraction protocol is a limitation of this study. These controls are meaningful for validating the extraction process and ensuring the reliability of the results. As their inclusion would have strengthened our findings, future studies are expected to address this limitation and provide more comprehensive validation of the results.

## Conclusions

It can be concluded that storage conditions like refrigeration for several days and ethanol preservation affect bovine fecal microbiota diversity in samples compared with freezing at -80˚C on day 0. Refrigeration for 3 and 7 days or ethanol preservation reduced the Shannon and Pielou evenness alpha diversity indices. However, for Faith PD and observed features, the four beta diversity indices included in this study showed that refrigeration at 4˚C for 3 days had a minimal effect on these measures with minimal impact on fecal microbiota diversity. When the refrigeration was for 7 days, or the samples were preserved in ethanol, there was a significant impact on the fecal microbiota diversity. Refrigeration for 7 days did affect microbiota diversity and is not recommended. Ethanol preservation is not recommended to replace immediate freezing to preserve fecal samples; however, this is an option for sample preservation when freezing is not possible for less than 3 days. Therefore, there is a tradeoff between the results obtained using storage and logical considerations, which are essential factors to keep in mind while designing large scale studies involving field and farm samples from remote locations.

## Acknowledgments

The authors would like to thank the farmers that participated in the study.

## Author Contributions

**Conceptualization:** Ana S. Jaramillo-Jaramillo, J. T. McClure, Henrik Stryhn, Kapil Tahlan, Javier Sanchez.

**Formal analysis:** Ana S. Jaramillo-Jaramillo, Henrik Stryhn, Javier Sanchez.

**Investigation:** Ana S. Jaramillo-Jaramillo, Henrik Stryhn, Javier Sanchez.

**Writing – original draft:** Ana S. Jaramillo-Jaramillo, J. T. McClure, Henrik Stryhn, Kapil Tahlan, Javier Sanchez.

**Writing – review & editing:** Ana S. Jaramillo-Jaramillo, J. T. McClure, Henrik Stryhn, Kapil Tahlan, Javier Sanchez.

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
