## [Decision Letter · Decision Letter 0]

21 May 2024

PONE-D-24-15864Effects of storage conditions on the microbiome of fecal samples collected from dairy cattlePLOS ONE

Dear Dr. Jaramillo-Jaramillo,

Thank you for submitting your manuscript to PLOS ONE. After careful consideration, we feel that it has merit but does not fully meet PLOS ONE’s publication criteria as it currently stands. Therefore, we invite you to submit a revised version of the manuscript that addresses the points raised during the review process.

We look forward to receiving your revised manuscript.

Kind regards,

Timothy Omara, PhD

Academic Editor

PLOS ONE

Journal Requirements:

"- Initials of the authors who received each award: JS

- Grant numbers awarded to each author: 1

- The full name of each funder: NSERC Discovery Grant (NSERC DG) 

- URL of each funder website: https://www.nserc-crsng.gc.ca/index_eng.asp

- Did the sponsors or funders play any role in the study design, data collection and analysis, decision to publish, or preparation of the manuscript? No"

Reviewers' comments:

Reviewer's Responses to Questions

**Comments to the Author**

1. Is the manuscript technically sound, and do the data support the conclusions?

Reviewer #1: Partly

Reviewer #2: Yes

2. Has the statistical analysis been performed appropriately and rigorously? 

Reviewer #1: Yes

Reviewer #2: Yes

3. Have the authors made all data underlying the findings in their manuscript fully available?

Reviewer #1: Yes

Reviewer #2: No

4. Is the manuscript presented in an intelligible fashion and written in standard English?

Reviewer #1: Yes

Reviewer #2: Yes

5. Review Comments to the Author

Reviewer #1: Statements and declarations:

Ethics: Field research was performed, but no permit details or authorisation was given in the section provided. Please include a statement if such permits/approvals were not required in your setting, or provide the approvals for the connected study for which the farms were enrolled.

Title:

In the last decade, efforts have been made to standardise microbiome terminology. Following these efforts, it would be more appropriate to refer to the “microbiota” instead of “microbiome” in this study, as the entire system and environment were not considered, but rather just the effect on the microbes themselves.

Introduction:

Lines 54 – 55: “Ribosomal” is more commonly used. Please do not omit articles like “the”, e.g., “the resistome”, “the mobilome”.

Methods:

Line 90: Was this temperature tested and proven to be stable for the length of time samples hung out, especially those from NS? It also matters whether the samples were directly on the ice packs or not. Depending on the quality of the box, we have found that temperatures can actually vary substantially, and could, arguably, have “worse” outcomes than having them at a semi-constant RT.

Line 95: Could you make it easier for the readers by specifying that it is the first 11 sample “sets” that arrived from the enrolled farms, and not individual samples?

Lines 99 – 108: Freezing the 3 day fridge subsample and not the 7 day fridge subsample has the potential to introduce bias, because the samples were treated differently – by one set going through a freeze-thaw and the other not. Freeze-thaw should be avoided in general, which is why a lot of stool extraction kits recommend not thawing frozen samples before extraction. I understand this was for batching purposes, but you need to keep in mind when analysing the data.

Lines 109 – 111: Why weren’t the 260/230 values considered? High fiber stools can often present difficulties for extractions and result in “dirty” DNA. While Illumina is more robust to this than some other chemistries, it is still a good measure of whether sequencing would have worked or not.

Lines 125 – 128: As far as I’m aware, GreenGenes was last updated in 2013. While QIIME2 often uses GG in their tutorials, in part because it is smaller and easier to compute, it would be hard to motivate to use such an outdated database as reference, for a paper up for review in 2024. I’d recommend switching to SILVA (there should be classifiers available for your version of QIIME2) or RDP.

While it is not incorrect, I would be wary of using the term OTU to describe biological taxonomic assignments, due to the potential confusion with the clustering outputs OTU vs ASV.

Line 143: Had to jump to results to check, but since the sequencing kit would have allowed up to 100 000 reads per sample I find it strange that the reads were so low per sample. Was this expected given results from other studies? The rarefaction depth is not “low” in the general sense, just unexpected given the potential output that the kit could have given, and the generally high bacterial load in the sample type used.

General methods comments: From past experience, it is easy to fall into this trap of not describing the sequencing and bioinformatics parts of the methods well enough. The methods here (especially in terms of sequencing pre-processing and analysis) really need more description for it to truly be reproducible. A lot of this could go in the supplementary methods to avoid clogging up the main text.

Some examples of what is missing:

Preprocessing: did you demultiplex, did you do any pre-QIIME2 QC, how was trimming of adapters/primers done? Which parameters were passed to DADA2? Which data did you take from QIIME2 to import and use in STATA, R, etc.? It is not clear that the taxonomic assignment was also done in QIIME.

What were your sequencing controls: Contamination is a huge issue in microbiome studies, so this needs to be addressed – you always need a negative(s)! Did you test your bioinformatics pipeline with a mock community control to ensure that it works properly?

Results:

Line 150: What do you mean with could not be sequenced? Did it fail library preparation or sequencing QC? How many samples were left in each group? Perhaps this information could be better represented in a table with the read ranges and things.

Lines 161 – 162: Figure labels 2B and 2C should be 1B and 1C.

Beta diversity: Would the immediate freezing not be better suited as a “control” to do comparisons against?

Table 1: Where is the adjusted p after Bonferroni correction column ?

Line 199: Relative abundance is not a good measure of taxonomic change on it’s own. Differential abundance, as compared to the “standard” of freezing immediately or another sample assigned as control, should be performed. Also think about investigating deeper than just Phylum level. Diff abundance can be done in QIIME2 (ANCOM, ANCOM-BC, gneiss) or R (various packages).

Discussion:

Line 249: Check the autocorrect typo for OTU.

Line 273 on: If F/B ratio is in the discussion, it should be introduced in the results in the taxonomy section first.

The discussion does not touch enough on WHY variability in results due to storage is a problem and should be countered where possible. Just one example is the fact that knowing that if certain microbes are affected more than others, it could mask or skew data, like which functions or resistance profiles can be detected or predicted in downstream analysis,.

Figures: It’s potentially just the version I received, but all the figures are a bit fuzzy. Why are figures 1 and 2 greyscale, when all the rest are in colour? It is harder to discern between groups with the greyscale.

Reviewer #2: Manuscript: Effects of storage conditions on the microbiome of fecal samples collected from dairy cattle

This study addresses a crucial and underexplored issue in bovine research: the effect of different storage conditions on the microbiota of dairy cattle. The aim was to investigate the effects of refrigeration at 4°C for three and seven days, as well as ethanol preservation, on the microbiota diversity of pooled fecal samples, with immediate freezing at -80°C after collection serving as the control. Establishing practical protocols for storing and transporting fecal samples from farm animals is essential, as researchers often face challenges in deciding the best strategies for collecting samples from farms located far from research laboratories. This study provides valuable information on the impact of refrigeration and ethanol preservation on fecal samples from cattle for microbiota studies.

Title: Change microbiome for microbiota and throughout the manuscript.

Introduction:

Line 46 to 62. This review can be condensed into a single paragraph (most information is already well known), concentrating on the knowledge gap that this study aims to address would be of importance.

Several studies in human and veterinary medicine (including dogs, cats, and horses) have evaluated the effect of different storage conditions on fecal and ruminal microbiota (e.g., Moossavi et al., BMC Microbiology, 19(1), 145; Granja-Salcedo et al., PLOS ONE, 12(4), e0176701; Barko et al., PLOS ONE, 19(2), e0294730). Acknowledging this previous work is recommended; however, it is even more important to clearly state how the present study differs from previous research to help the reader understand its significance. For instance, a previous study (Song SJ, mSystems. 2016;1(3):e00021-16) strongly recommend against 75% ethanol for preservation of microbiota, so here in the introduction the authors can hypothesise whether in cattle 75% ethanol would perform similar or different.

Line 77: Additionally, the introduction of the manuscript fails to explain the importance of establishing a practical storage and transportation method for pooled sampling in bovine research. Also explain when pooled samples could be (or are) used in bovine microbiota research

Line 77: any hypothesis?

M&M

Line 124: Was the location of the farm considered in the statistical models for alpha- and beta-diversity? Samples from NS were kept refrigerated below 8°C for 24 hours, while those from PEI were kept refrigerated below 8°C for only 3 hours. Studies in horses and dogs have shown changes in fecal microbiota after 6 hours of temperature exposure. Could the storage duration before processing have affected the microbiota analysis?

Results:

Line 161: this sentence appears to be incomplete.

Line 172: What is the meaning of largest discrepancies with less variability? Restate please

Line 190: Please discuss why the effect of storage condition was different for each of the b-diversity measures used?

Line 199: Please explain why the taxonomy analysis was performed only at the phylum level and not at the genus level. In this study, the V6-V8 region of the 16S rRNA gene was amplified, yielding DNA fragments of approximately 400-500 bp. This length allows for genus-level analysis, which could determine which genera within each phylum are affected by the storage conditions.

Discussion

Line 225: any reference supporting this statement? Also, which bacteria are more likely do increase or decrease with the type of storage and preservation method used. do the results of the present study support previous findings?

Line 227: what about in species with similar diets and likely similar gut microbiota?

Line 235: “significant varied”? increase or decrease?

Line 236: any idea why?

Line 261: which species? All or in which species has been the effect of refrigeration being investigated?

Line 262 – 268: in practical terms and based on your general objective, is it Ethanol 70% recommended or no for preservation of bacterial DNA?

Line 269 – 272: is there any studies in calves or adult cattle showing the effect of farm on the fecal microbiota?

Any study limitations?

Figure 1 and 2: consider using color in the figure.

Are the dataset Publicly available? If so, provide information about where and project number.

6. PLOS authors have the option to publish the peer review history of their article (what does this mean?). If published, this will include your full peer review and any attached files.

Reviewer #1: No

Reviewer #2: **Yes: **Diego Gomez

---

## [Author Response · Author response to Decision Letter 0]

6 Jun 2024

June 5, 2024

Journal: PLOS ONE

Manuscript ID: PONE-D-24-15864

Title: "Effects of storage conditions on the microbiota of fecal samples collected from dairy cattle" 

Thank you for giving us the opportunity to submit a revised draft of our manuscript titled "Effects of storage conditions on the microbiota of fecal samples collected from dairy cattle" to PLOS ONE. We appreciate the time and effort that you and the reviewers have dedicated to providing valuable feedback on our manuscript. We are grateful to the reviewers for their insightful comments on our paper. We have now been able to incorporate changes to reflect the suggestions provided by the reviewers, as indicated in the 'bullet points' below. 

• We ensured that our manuscript meets PLOS ONE's style requirements.

"- Initials of the authors who received each award: JS

- Grant numbers awarded to each author: 1

- The full name of each funder: NSERC Discovery Grant (NSERC DG) 

- URL of each funder website: https://www.nserc-crsng.gc.ca/index_eng.asp

- Did the sponsors or funders play any role in the study design, data collection and analysis, decision to publish, or preparation of the manuscript? No"

 

Reviewer #1

Main comments: 

Statements and declarations:

Ethics: 

Field research was performed, but no permit details or authorisation was given in the section provided. Please include a statement if such permits/approvals were not required in your setting, or provide the approvals for the connected study for which the farms were enrolled.

• As suggested by the reviewer, an Ethics statement was added to the Materials and Methods section, where we specified that the owners of the farms selected for conducting this work gave their permission to carry out the study and no additional permissions for sample collection and analysis were needed (Line 85).

Title: 

In the last decade, efforts have been made to standardise microbiome terminology. Following these efforts, it would be more appropriate to refer to the "microbiota" instead of "microbiome" in this study, as the entire system and environment were not considered, but rather just the effect on the microbes themselves.

• We agreed with your suggestion and given that our study focused primarily on the effects on the microbial community itself, it is indeed more precise to use the term "microbiota." The adjustments were made in the title and throughout the manuscript when applicable.

Introduction:

Lines 54 – 55: "Ribosomal" is more commonly used. Please do not omit articles like "the", e.g., "the resistome", "the mobilome".

• We corrected "ribosome" to "ribosomal" and added the missing articles as suggested (Line 51).

Methods:

Line 90: Was this temperature tested and proven to be stable for the length of time samples hung out, especially those from NS? It also matters whether the samples were directly on the ice packs or not. Depending on the quality of the box, we have found that temperatures can actually vary substantially, and could, arguably, have "worse" outcomes than having them at a semi-constant RT.

• The location of the farm and the storage duration before processing indeed have the potential to affect the microbiota analysis. While we did not directly control the stability of a constant temperature throughout the sample storage duration, we adhered to our sampling protocols to minimize variability including avoiding direct contact of the samples with ice packs.

• To address this variability, we included the farm effect as a random effect in all alpha- and beta-diversity statistical models. This approach was intended to account for potential variability and bias related to transportation time and temperature.

• Specifically for the NS samples, controlling these aspects was particularly challenging. However, our statistical approach helps mitigate these issues and provides a more accurate representation of the microbiota under the given storage conditions (Line 89-101).

Line 95: Could you make it easier for the readers by specifying that it is the first 11 sample "sets" that arrived from the enrolled farms, and not individual samples?

• As suggested by the reviewer, it was specified that we took sample sets made of five individual samples per farm and that the first 22 sample sets that arrived at the laboratory were processed (Line 104). 

Lines 99 – 108: Freezing the 3 day fridge subsample and not the 7 day fridge subsample has the potential to introduce bias, because the samples were treated differently – by one set going through a freeze-thaw and the other not. Freeze-thaw should be avoided in general, which is why a lot of stool extraction kits recommend not thawing frozen samples before extraction. I understand this was for batching purposes, but you need to keep in mind when analysing the data.

• As many stool extraction kits recommend, we agree that freeze-thaw cycles should generally be avoided to prevent potential bias. However, considering the necessity of a seven-day waiting period, we aimed to minimize the overall handling time for the 7-day samples by processing them immediately after this period. This approach was taken to avoid adding extra time to the already frozen samples. We also operated under the assumption that freezing samples at -80°C would halt most microbial activity until DNA extraction, thus mitigating potential bias. Additionally, we ensured that no more than one freeze-thaw cycle occurred before extraction (Line 108-113).

Lines 109 – 111: Why weren't the 260/230 values considered? High fiber stools can often present difficulties for extractions and result in "dirty" DNA. While Illumina is more robust to this than some other chemistries, it is still a good measure of whether sequencing would have worked or not.

• We agree that 260/230 values are important, especially in high-fiber stool samples, as they can indicate the quality of the DNA extraction. In our study, we measured the 260/230 values to assess DNA quality in our laboratory. Additionally, the external laboratory re-evaluated these values before proceeding with sequencing to ensure the suitability of the samples. This dual-check process helped us confirm the integrity of the DNA for sequencing. The 260/230 ratio was added to the text to specify that these values were considered as well (Line 115-121).

Lines 125 – 128: As far as I'm aware, GreenGenes was last updated in 2013. While QIIME2 often uses GG in their tutorials, in part because it is smaller and easier to compute, it would be hard to motivate to use such an outdated database as reference, for a paper up for review in 2024. I'd recommend switching to SILVA (there should be classifiers available for your version of QIIME2) or RDP.

• We acknowledge that using a more current database is crucial for accuracy and relevance in 2024 publications. However, far from fully characterizing and describing the microbiota, our objective in the present study was to compare the effect of preservation methods on the results. Due to resource and practicality considerations, we decided to use a database with lower CPU time and memory requirements. Based on your recommendation, we will consider switching to the SILVA or RDP databases in the future, which should be compatible with our version of QIIME2 (Line 123-135).

While it is not incorrect, I would be wary of using the term OTU to describe biological taxonomic assignments, due to the potential confusion with the clustering outputs OTU vs ASV.

• To avoid confusion, we replaced the term OTU with the plural OTUs to describe the taxonomical units throughout the manuscript when applicable.

Line 143: Had to jump to results to check, but since the sequencing kit would have allowed up to 100 000 reads per sample I find it strange that the reads were so low per sample. Was this expected given results from other studies? The rarefaction depth is not "low" in the general sense, just unexpected given the potential output that the kit could have given, and the generally high bacterial load in the sample type used.

• Even though the previous studies we referenced describe similar number of reads per fecal sample (Vasco et al., 2021), the actual number of reads obtained can be influenced by several factors, including DNA quality, sample preparation, and inherent variability in sequencing runs. 

• Our rarefaction depth was selected based on a balance between capturing sufficient diversity and ensuring consistency across samples (Line 154). 

• Full reference: Vasco, K., Nohomovich, B., Singh, P., Venegas-Vargas, C., Mosci, R. E., Rust, S., Bartlett, P., Norby, B., Grooms, D., Zhang, L., & Manning, S. D. (2021). Characterizing the Cattle Gut Microbiome in Farms with a High and Low Prevalence of Shiga Toxin Producing Escherichia coli. Microorganisms, 9(8), Article 8. https://doi.org/10.3390/microorganisms9081737

General methods comments: From past experience, it is easy to fall into this trap of not describing the sequencing and bioinformatics parts of the methods well enough. The methods here (especially in terms of sequencing pre-processing and analysis) really need more description for it to truly be reproducible. A lot of this could go in the supplementary methods to avoid clogging up the main text.

Some examples of what is missing:

Preprocessing: did you demultiplex, did you do any pre-QIIME2 QC, how was trimming of adapters/primers done? Which parameters were passed to DADA2? Which data did you take from QIIME2 to import and use in STATA, R, etc.? It is not clear that the taxonomic assignment was also done in QIIME.

• Thank you for your feedback on the methods section. We understand the importance of providing detailed descriptions to ensure reproducibility. While the sequencing was conducted by an external laboratory, and we assumed they followed the correct protocols, we unfortunately do not have access to the specific details of those protocols. Our primary objective in this study was to investigate the effects of different conservation methods on microbiota analysis, and our expertise is not in bioinformatics. Therefore, we did not consider it necessary to include more bioinformatics details in the main text.

• However, to address your specific questions:

- Yes, we performed demultiplexing.

- No, we did not conduct any pre-QIIME2 quality control (QC).

- We took the features table, the taxonomy, the unrooted tree, and the metadata from QIIME2 and imported them into R for further analysis using the phyloseq package.

- The taxonomic assignment was done in QIIME2.

What were your sequencing controls: Contamination is a huge issue in microbiome studies, so this needs to be addressed – you always need a negative(s)! Did you test your bioinformatics pipeline with a mock community control to ensure that it works properly?

• Thank you for your important question regarding sequencing controls. In this study, we sent the DNA samples to an external laboratory for sequencing, and we assumed they implemented the necessary negative controls during the PCR process to address potential contamination.

• Regarding the inclusion of a mock community control, we did not include such a control in our study. Our primary focus was not on the deep characterization of the microbiota but rather on comparing the effects of different preservation methods on the obtained results. While a mock community control can be valuable for validating the bioinformatics pipeline, our study aimed to highlight relative differences between storage conditions rather than provide an exhaustive microbiota profile.

Results:

Line 150: What do you mean with could not be sequenced? Did it fail library preparation or sequencing QC? How many samples were left in each group? Perhaps this information could be better represented in a table with the read ranges and things.

• The mentioned samples failed the sequencing quality control (QC) in this case. The total number of samples included in the final analysis was 71. As suggested, we added a table to the manuscript with the number of samples left in each group, the average reads per sample, and their ranges for a more straightforward interpretation (Line 162-166).

Lines 161 – 162: Figure labels 2B and 2C should be 1B and 1C.

• The mentioned labels regarding the results on Figure 1 were corrected as suggested by the reviewer (Line 171-178).

Beta diversity: Would the immediate freezing not be better suited as a "control" to do comparisons against?

• In this analysis, our objective was to compare every conservation method to each other and assess the distances between each pair of methods, as illustrated in Figure 2. Therefore, we did not include a "control" or a "golden standard" method. Our goal was to comprehensively compare all conservation methods rather than benchmark against a single standard.

• We noted that the objective of this study was written in a confusing way that suggested we use the immediate freezing method as a standard for comparison. We reworded this paragraph, specifying that we aimed to compare every conservation method to the others. 

Table 1: Where is the adjusted p after Bonferroni correction column?

• The reference to the Bonferroni adjustment in the beta diversity results was a mistake. Table 1 summarizes only the PERMANOVA results, where the p-values are already adjusted. After conducting the general linear mixed models, the Bonferroni adjustment was applied for alpha diversity pairwise comparisons. We corrected the paragraph and changed the table's position to make it easier for the reader to interpret the information it presents (Line 194 and Table 2).

Line 199: Relative abundance is not a good measure of taxonomic change on it's own. Differential abundance, as compared to the "standard" of freezing immediately or another sample assigned as control, should be performed. Also think about investigating deeper than just Phylum level. Diff abundance can be done in QIIME2 (ANCOM, ANCOM-BC, gneiss) or R (various packages).

• Line 213

• In our study, the taxonomy component has a descriptive purpose and aims to show the general effects of different conservation methods on bacterial phyla. We chose to use relative abundance as a straightforward measure to illustrate changes when comparing each method with the others. Additionally, as we did not want to characterize the fecal microbiota deeply but only to catch possible effects of the included conservation methods, we kept the phylum level as a general picture of the taxonomy classification made to the microbiota of the samples. 

• As mentioned, we did not include a "control" or "golden standard" method. Our goal was to comprehensively compare all conservation methods rather than benchmark against a single standard. While we acknowledge that relative abundance alone may not fully capture taxonomic changes, we believe it effectively highlights the overall trends.

• However, we appreciate your suggestion regarding differential abundance analysis and deeper taxonomic investigation. These approaches could provide more detailed insights and will be considered for future studies. 

• We reworded the paragraph of the taxonomy component to make it easier to understand.

Discussion:

Line 249: Check the autocorrect typo for OTU.

• We checked and corrected the word "OUT" to "OTUs” (Line 269).

Line 273 on: If F/B ratio is in the discussion, it should be introduced in the results in the taxonomy section first.

• We agree that the F/B (Firmicute

---

## [Decision Letter · Decision Letter 1]

26 Jun 2024

PONE-D-24-15864R1Effects of storage conditions on the microbiota of fecal samples collected from dairy cattlePLOS ONE

Dear Dr. Jaramillo-Jaramillo,

Thank you for submitting your manuscript to PLOS ONE. After careful consideration, we feel that it has merit but does not fully meet PLOS ONE’s publication criteria as it currently stands. Therefore, we invite you to submit a revised version of the manuscript that addresses the points raised during the review process.

We look forward to receiving your revised manuscript.

Kind regards,

Timothy Omara, PhD

Academic Editor

PLOS ONE

Journal Requirements:

Reviewers' comments:

Reviewer's Responses to Questions

**Comments to the Author**

1. If the authors have adequately addressed your comments raised in a previous round of review and you feel that this manuscript is now acceptable for publication, you may indicate that here to bypass the “Comments to the Author” section, enter your conflict of interest statement in the “Confidential to Editor” section, and submit your "Accept" recommendation.

Reviewer #1: (No Response)

Reviewer #2: All comments have been addressed

2. Is the manuscript technically sound, and do the data support the conclusions?

Reviewer #1: Yes

Reviewer #2: Yes

3. Has the statistical analysis been performed appropriately and rigorously? 

Reviewer #1: Yes

Reviewer #2: Yes

4. Have the authors made all data underlying the findings in their manuscript fully available?

Reviewer #1: Yes

Reviewer #2: Yes

5. Is the manuscript presented in an intelligible fashion and written in standard English?

Reviewer #1: Yes

Reviewer #2: Yes

6. Review Comments to the Author

**Reviewer #1:** I find it odd that a group of authors' who state that their expertise is not bioinformatics, would undertake to write a manuscript based almost entirely off of bioinformatics analyses, and not bring onboard an expert? It is not that the analysis was poorly done, it is just that you should not use not being an expert as an rebuttal to leave out details needed for reproducibility of the protocol.

For future studies, it is the responsibility of the authors to ensure that they are familiar with the workflow of the sequencing provider and that correct procedure is followed - from experience one cannot assume that providers would do the "proper" things. Surely it would have been possible to have reached out and asked the provider what kind of controls they had instead of rebutting saying you assume they did?

Lastly, while the focus of the study was not to present a robust microbiota profile, I would argue that in order to properly represent the changes happening as a result of the conditions the samples were exposed to, other experimental factors such as contamination (which could skew results) should have been been controlled for, and the data QC and taxonomy assignment be as robust as possible. Differential abundance is also not hard to do in QIIME2, and could have easily been applied to see which, if any, phyla are differentially abundant and therefore the main drivers of changes due tot the conditions.

As adding controls is clearly not possible at this stage, please at least add a statement indicating that extraction negative controls and positive controls were not included in the protocol and please consider performing more robust analyses in future. It would only serve to strengthen the results you see, regardless of the focus.

I accept the remainder of the changes and/or rebuttals to my comments.

**Reviewer #2:** Thanks to the authors for addressing the reviewer's concerns. This reviewer does not have additional comments.

7. PLOS authors have the option to publish the peer review history of their article (what does this mean?). If published, this will include your full peer review and any attached files.

Reviewer #1: No

Reviewer #2: No

---

## [Author Response · Author response to Decision Letter 1]

24 Jul 2024

Thank you for giving us the opportunity to submit a second revised draft of our manuscript titled "Effects of storage conditions on the microbiota of fecal samples collected from dairy cattle" to PLOS ONE. We appreciate the time and effort that you and the reviewers have dedicated to providing valuable feedback on our manuscript. We are grateful to the reviewers for their insightful comments on our paper. We have now been able to incorporate changes to reflect the suggestions provided by the reviewers.

---

## [Editor Report · Decision Letter 2]

26 Jul 2024

Effects of storage conditions on the microbiota of fecal samples collected from dairy cattle

PONE-D-24-15864R2

Dear Dr. Jaramillo-Jaramillo,

We’re pleased to inform you that your manuscript has been judged scientifically suitable for publication and will be formally accepted for publication once it meets all outstanding technical requirements.

Kind regards,

Timothy Omara, PhD

Academic Editor

PLOS ONE
---

## [Editor Report · Acceptance letter]

1 Aug 2024

PONE-D-24-15864R2 

PLOS ONE

Dear Dr. Jaramillo-Jaramillo, 

I'm pleased to inform you that your manuscript has been deemed suitable for publication in PLOS ONE. Congratulations! Your manuscript is now being handed over to our production team.

Kind regards, 

on behalf of

Dr. Timothy Omara 

Academic Editor

PLOS ONE